# Emotion Regulation and Self-Efficacy: The Mediating Role of Emotional Stability and Extraversion in Adolescence

**DOI:** 10.3390/bs14030206

**Published:** 2024-03-04

**Authors:** Pablo Doménech, Ana M. Tur-Porcar, Vicenta Mestre-Escrivá

**Affiliations:** 1Faculty of Education, Valencian International University (VIU), Pintor Sorolla, 21, 46002 Valencia, Spain; pablo.domenech@campusviu.es; 2Faculty of Psychology, University of Valencia, Avda. Blasco Ibáñez, 21, 46010 Valencia, Spain; maria.v.mestre@uv.es

**Keywords:** emotional self-efficacy, emotion regulation, extraversion, emotional stability, adolescence

## Abstract

The feeling of emotional self-efficacy helps people understand how to handle positive and negative emotions. Emotion regulation is the process that helps people control their emotions so that they can adapt to the demands of the environment. This study has a twofold aim. First, it examines the relationships among emotion regulation, the personality traits of extraversion and emotional stability, and the feeling of emotional self-efficacy for positive and negative emotions in an adolescent population. Second, it examines the mediating role of personality traits (extraversion and emotional stability) in the relationship between emotion regulation and emotional self-efficacy for positive and negative emotions. The participants were 703 adolescents (49.9% male and 50.1% female) aged between 15 and 18 years (*M* = 15.86, *SD* = 0.30). Significant relationships were observed among emotion regulation, the personality traits of extraversion and emotional stability, and emotional self-efficacy for positive and negative emotions. The structural equation model confirmed the direct link between emotion regulation and emotional self-efficacy and mediation by the personality traits of extraversion and emotional stability. This study confirms that emotional self-efficacy is connected to the emotion regulation strategies that adolescents use. Effective emotion regulation encourages self-perception and emotional coping. The results are discussed in connection to previous research.

## 1. Introduction

In everyday life, people must deal with emotionally charged situations that can cause stress and a lack of efficacy in decision making. These situations have a particularly strong impact in adolescence, a period of physiological and associated psychological, emotional, and behavioural changes [1]. These changes create instability and variations in personality traits, even potentially evolving towards a negative personality [2]. Such changes can occur in the process of transition into adulthood and can stay with people for the rest of their lives. Therefore, the ability of adolescents to manage everyday situations effectively and achieve the desired outcome will help them in the process of self-management (as agentic beings) to ensure the sound development of beliefs of emotional efficacy [3]. This emotional self-efficacy can influence the types of goals people set for themselves and even their professions as adults [4]. In addition, personality traits play a role in the way in which people deal with problems. Extraversion and emotional stability (or conversely neuroticism) are the two personality traits that most greatly enable adaptation to the demands of the environment [5] and that share the strongest relationship with emotional self-efficacy [6].

In this context, it is considered that emotion regulation, as a process that helps control emotions, will play a key role. Individuals who are able to regulate their emotional state will mature in a more adaptive manner in their environment. Overall, it will increase their degree of personal and social well-being [7]. Consequently, it could be instructive to conduct an analysis of these aspects during adolescence, a period of change and instability in personality traits. Specifically, an analysis of the relationships between the ability for emotion regulation and the development of feelings of emotional self-efficacy, as well as the mediating role of the personality traits of emotional stability and extraversion, can provide insight into the importance of encouraging the development of an emotionally stable personality and the ability to externalise feelings and emotions effectively. This process can have benefits in terms of interactions with others, given that the belief of being effective in emotion management can play a role [8,9].

Studies have shown the direct relationships between self-efficacy and positive and negative emotion regulation strategies [10]. However, the mediating role of emotional stability and extraversion in this relationship remains unknown. The present study attempts to fill this gap. The results provide insight into the importance of encouraging strategies aimed at developing these two personality traits in adolescence, a period of change that leads to a degree of immaturity in personality traits [11,12] before these traits stabilise and remain throughout adulthood [13]. Therefore, the aim of this study is to analyse the role of emotion regulation (cognitive reappraisal and expressive suppression) as a predictor of self-efficacy for positive and negative emotions, as well as the mediating role of the personality traits of emotional stability and extraversion.

### 1.1. Personality and Emotional Self-Efficacy

Personality traits are defined as relatively stable patterns of thought, feelings, and behaviour in an individual [14]. However, in early adolescence, an individual’s personality becomes unstable [15]. Personality traits change in the opposite way, becoming less mature until adulthood, when they mature again [2,13,16,17]. Adolescence is characterised by changes in personality traits [11,12]. As a result, behavioural patterns and forms of temperament gradually establish themselves in relation to the environment [18].

Social cognitive theory suggests that personality traits are shaped by multiple processes. These processes include the generalisation of skills and self-awareness, which form through interactions between a person and the environment, as well as transitions in life. These traits ultimately become behavioural patterns [19]. The five-factor model refers to five big traits: extraversion, agreeableness, conscientiousness, neuroticism (vs. emotional stability), and openness to experience. There are also superficial traits or expressions of personality referring to people’s beliefs, abilities, values, and attitudes. These values depend on core structures, but they are flexible and can be shaped by environmental influences [20]. This superficial group includes self-evaluations of efficacy, which help create awareness of how to manage positive and negative emotions [8].

Two of the big five personality traits, extraversion and neuroticism (vs. emotional stability), predict positive and negative emotional states [21]. In this model, neuroticism is the opposite of emotional stability. Extraversion and emotional stability enable adaptation to the demands of the environment [5]. Extroverts tend to have positive emotional states and are satisfied with life. In contrast, neurotic people tend to be in a negative emotional state, among other reasons because they focus on negativity, which makes them more prone to stimuli that cause negative emotions [22]. Also, people with high levels of emotional stability have more resources to tackle negative emotional states [23,24].

Emotional stability and extraversion tend to encourage people to seek a greater connection to their environment. People with high levels of emotional stability have resources to relate to others via assertive attitudes that help them defend their needs. Extroverts seek social interactions and others with whom to relate [19,25]. They take an interest in their company and tend to be assertive [26]. Thus, both of these personality traits can play a crucial role in social relations and in the search for solutions to day-to-day events that lead to tension, which are widely present in adolescents’ academic and occupational environments [5].

Emotional self-efficacy is defined as people’s belief in their capacity to achieve proposed goals efficiently and achieve desired outcomes [27]. Social cognitive theory draws on an agentic perspective, in which agency works through a process of causality, involving personal, behavioural, and environmental factors [3]. In this process, people actively shape the course of their lives. Hence, they can anticipate the consequences of their actions to guide them towards plans of action, supported by a process of reflection and self-regulation [27]. Achieving goals largely depends on self-efficacy, which defines the course of action in a process of self-management of psychosocial functioning [28]. This self-efficacy also influences motivation and perseverance in the face of difficulties, as well as the expectations of outcomes [29]. Beliefs and the process of self-regulation also influence the sense of achievement or failure [3]. While experiencing achievement, a person’s feeling of self-efficacy grows, while motivation, capacity, and interest in the task all improve too. This occurs via cognitive, motivational, affective, and decision-making mechanisms [27]. The experience of failure may have a negative impact on the feeling of self-efficacy and may reduce interest and motivation to achieve proposed goals [29].

The feeling of efficacy that accompanies achievement or failure Is associated with positive and negative emotions. Self-efficacy for positive emotions refers to a person’s perceived ability to express emotions such as happiness, enthusiasm, and pride as a result of success and pleasurable events. Self-efficacy for negative emotions refers to a person’s perceived capacity to handle and improve emotions such as anger, distress, irritation, or dejection, expecting negative results [1]. Authors have noted the need for further research into connections between personality traits, emotional self-efficacy, and emotion regulation [8].

Furthermore, the different strategies for handling emotions are related to personality traits [30] and emotional self-efficacy [6]. Studies have shown that self-efficacy and emotion regulation link certain personality traits such as extraversion and neuroticism (vs. emotional stability) to a better quality of life because they are associated with positive emotionality and more active and dynamic procedures [31].

### 1.2. Emotion Regulation

Emotion regulation is a process that helps people control their emotions to adapt to the demands of their environment [32]. It involves handling positive and negative emotions [33]. The process model of emotion regulation [34,35] divides the strategies of emotion regulation into two groups: one focusing on antecedents of the emotional experience and another on responses [33,34,35]. Antecedent-focused strategies are activated before the emotional experience occurs, such as cognitive reappraisal. In contrast, response-focused strategies are to those that are activated once the emotional process has begun, such as expressive suppression [9].

Cognitive reappraisal is considered an adaptive strategy linked to better social relations [7], better interpersonal functioning [9,36], and higher levels of positive affect [37]. Conversely, expressive suppression is considered a maladaptive strategy related to greater difficulties in establishing social relations and a lower level of well-being [38,39,40]. Some studies of adolescence have linked expressive suppression to emotional dysregulation and cognitive reappraisal to adaptive emotional coping [40]. Other studies have linked high scores in expressive suppression and low scores in cognitive reappraisal to problematic types of behaviour in an academic environment [41]. Despite these results, expressive suppression can be adaptive in preschool environments [42], even though it becomes maladaptive in the long term [43].

### 1.3. Emotion Regulation, Emotional Self-Efficacy, and Personality

Empirical evidence has confirmed the connections between emotional self-efficacy and emotion regulation. It has been observed that emotional self-efficacy is related positively to cognitive reappraisal but negatively to expressive suppression [10]. Moreover, the way in which self-efficacy is evaluated is important in understanding the process of emotion regulation [44,45]. Bujor and Turliuc [46] found that, in addition to reducing the intensity of negative emotions, cognitive reappraisal also boosted positive emotions, whereas expressive suppression could be an efficient way of regulating the expression but not the experience of emotions. High scores in expressive suppression are associated with greater negative affect and lower positive affect [38,47,48]. Emotional self-efficacy refers to individuals’ perceptions of their emotional management, while emotion regulation refers to how individuals manage their emotions [8,9]. Although these variables are both related to emotions, self-efficacy is a belief about emotional management, whereas regulation is the way individuals act when faced with their emotions. Emotional self-efficacy distinguishes between positive and negative emotions. In contrast, emotion regulation focuses on the strategies used to manage those emotions. Emotion regulation focuses on cognitive reappraisal and expressive suppression strategies, while emotional self-efficacy focuses on emotional management beliefs regarding positive and negative emotions. Therefore, it is important to understand the relationship between emotional self-efficacy and emotion regulation [8].

Research into relationships between personality traits and emotional self-efficacy has confirmed that high scores in self-efficacy are related to high scores in extraversion and low scores in neuroticism [6]. People with higher scores in neuroticism (or low scores in emotional stability) are easily irritable and respond unsuitably to stressful factors of negative affect [49]. Furthermore, they have difficulties in regulating negative emotions [50]. Neuroticism is strongly and negatively correlated with self-efficacy in coping with negative emotions of anger, irritation, despair, and distress, although it does not appear to have such a strong link to self-efficacy for positive emotions [45].

In addition, high levels of emotional stability are related to self-efficacy for negative as well as positive emotions [51]. People with high emotional stability tend to experience negative emotions of anger, sadness, and emotional distress at lower levels than those with low emotional stability [46]. Furthermore, emotional stability is positively correlated with self-efficacy for the negative emotions of despair and distress [52]. Meanwhile, high scores in extraversion are associated with experiences of positive affect [53]. For instance, Shi et al. [52] found a significant positive correlation between extraversion and self-efficacy for positive emotions.

## 2. Aims and Hypotheses

This study has a twofold aim: (i) to analyse the relationships among emotional self-efficacy for positive and negative emotions, emotion regulation, and the personality traits of extraversion and emotional stability in an adolescent population; and (ii) to verify the mediating role of these two personality traits in relation to emotion regulation and self-efficacy for positive and negative emotions, following the suggestions of Caprara et al. [8]. The goal is thus to observe whether extraversion and emotional stability mediate the relationship between emotion regulation and emotional self-efficacy.

Extraversion and neuroticism were included as mediating variables for two reasons. First, extraversion and neuroticism have direct effects on positive and negative affect, whereas openness, conscientiousness, and agreeableness may have an instrumental (indirect) effect on positive and negative affect [54]. These findings from the literature justify the use of these two variables as mediating variables [54]. Additionally, neuroticism increases during adolescence, especially among girls [2,16]. Moreover, previous studies have used extraversion and neuroticism as mediating variables [55,56,57]. Second, personality variables are unstable during adolescence [12,15], as can also be observed in our research. They have been found to have standard deviations of 1.28 and 1.20. Thus, if they present variability, they cannot be considered constants. Based on the previous findings and a review of the literature, it is hypothesised that the following relationships will hold in adolescence:

**H1.** 
*Positive relationships will be observed among cognitive reappraisal, the personality traits of emotional stability and extraversion, and self-efficacy. Cognitive reappraisal will be related to self-efficacy for both positive and negative emotions.*


**H2.** 
*Expressive suppression will be negatively related to positive emotional self-efficacy and positively related to self-efficacy for negative emotions.*


**H3.** 
*Emotion regulation will influence self-efficacy of positive and negative emotions (despair and distress; anger and irritation).*


**H4.** 
*The personality factors of extraversion and emotional stability will act as mediating variables between emotion regulation and self-efficacy for positive and negative emotions (despair and distress; anger and irritation).*


## 3. Methods

### 3.1. Participants

The participants were 703 adolescents (49.9% male and 50.1% female) aged between 15 and 18 years (*M* = 15.86, *SD* = 0.30). The age distribution of the sample was as follows: 35.7% were 15 years old, 46.2% were 16 years old, 13.9% were 17 years old, and 4.1% were 18 years old. All the adolescents were studying either in the last year of compulsory secondary education (49.6%) or in the post-16 baccalaureate (50.4%) in the Spanish province of Valencia. Finally, 49.5% studied in state schools, and 50.5% studied in private or subsidised schools.

### 3.2. Instruments

Emotion Regulation Questionnaire (ERQ) [9] (Spanish adaptation by [36]).

This instrument evaluated the two strategies of emotion regulation: cognitive reappraisal and expressive suppression. A seven-point Likert scale was used, ranging from 1 (*completely disagree*) to 7 (*completely agree*). Cognitive reappraisal had a Cronbach’s alpha score of 0.80, and expressive suppression had a Cronbach’s alpha score of 0.74. These values were similar to those for the Spanish validation (α = 0.79 for cognitive reappraisal and α = 0.75 for expressive suppression) [36]. The instrument had 10 items. An example item was “When I want to feel the most positive emotion (such as joy or fun), can I change what I am thinking?” and “I keep my emotions to myself”.

Ten-Item Personality Inventory, TIPI [57].

This brief measuring instrument was used for the big five domains of personality traits [57]. It evaluated the personality traits of extraversion (enthusiastic), agreeableness (affectionate), conscientiousness (reliable and self-disciplined), emotional stability (not easily irritated), and openness to experiences (inquisitive). A seven-point Likert scale was used, ranging from 1 (*completely disagree*) to 7 (*completely agree*). It consisted of 10 items. An example item was “Calm, emotionally stable” for emotional stability and “Extraverted, enthusiastic” for extraversion. This questionnaire has been widely accepted for use in scientific research and has provided results comparable to other more extensive personality questionnaires [57,58]. It was conceptualised in terms of the domains of behaviour from behaviour domain theory (BDT), estimating the constructs by inference based on generalisation in the population. The items for each trait were understood to correspond to behaviour, and their responses did not have to be interrelated [59]. Short personality questionnaires, which follow the five-factor model, have been shown to have adequate psychometric criteria and tend to be used for reasons of economy of effort [60]. The Cronbach’s alpha scores were 0.7 for extraversion and 0.67 for neuroticism (vs. emotional stability). Alpha scores greater than 0.60 can be considered acceptable [61,62,63]. The item–total (item–item) correlation coefficient was *r* = 0.544 ** for extraversion and *r* = 0.507 ** for emotional stability [64]. Given the variable nature of personality traits during adolescence, the indications of Wu and Zumbo [65] were followed in this study. Accordingly, the variables of emotional stability and extraversion were included as mediating variables.

Regulatory Emotional Self-Efficacy, RESE [8].

This scale examined perceived self-efficacy, geared towards showing or experiencing positive or negative affect. The scale had 12 items, using a five-point Likert scale ranging from 1 (*incapable*) to 5 (*completely capable*). This scale had two subdimensions: (1) emotional self-efficacy perceived to express positive affect (POS), with alpha = 0.74; and (2) emotional self-efficacy perceived to express negative affect. Negative affect had two further subfactors: (1) emotional self-efficacy perceived in handling anger and irritation (ANG), with alpha = 0.72; and (2) self-efficacy perceived in handling despair and distress (DES), with alpha = 0.74. An example item was “How well can you: Express joy when good things happen to you?”

### 3.3. Procedure

This study was cross-sectional. The sample was selected based on the classification of secondary schools in the Spanish province of Valencia. The project received permission from the schools and support from the teaching staff. Group evaluations were carried out during class time in the adolescents’ classrooms. Two expert evaluators supervised the entire procedure. For participation, authorisation from the schools, families, and students themselves was first received. The parents and students completed and signed the informed consent form. The management team of the schools approved the intervention. The ethical guidelines of the Helsinki Declaration for this type of research were followed. Participation was voluntary, anonymous, and confidential. At any moment in the process, the students could withdraw their participation if they wished. However, no student decided to withdraw from the evaluation. The evaluations were carried out in sessions of approximately 45 min. The data were collected between February and May 2019.

### 3.4. Data Analysis

First, to provide background on the variables, the descriptive statistics and Pearson correlations were examined for all variables. Second, a predictive structural equation model was estimated using confirmatory techniques. The dimensions of emotion regulation (cognitive reappraisal and expressive suppression) acted as antecedent variables. The emotional self-efficacy dimensions (POS, ANG, and DES) acted as consequent variables. The personality variables of emotional stability and extraversion acted as mediator variables. The statistical analyses were carried out in SPSS 24.0 and Mplus 8 [66].

## 4. Results

### 4.1. Descriptive and Correlation Analyses

Table 1 shows a platykurtic (negative kurtosis) distribution for all variables. The variables of expressive suppression, DES and ANG, had positive asymmetry, with longer right tails. The other variables had negative asymmetry, with longer left tails.

Table 2 shows the correlations between the variables. The strategies of cognitive reappraisal were positively correlated with extraversion, emotional stability, self-efficacy for positive emotions, and self-efficacy for negative emotions (DES and ANG). The correlation indices with DES (*r* = 0.408 **, *p* < 0.01) and with ANG (*r* = 0.422 **, *p* < 0.01) were high. Expressive suppression was negatively correlated with extraversion and positive emotions. However, the relationships with emotional stability and with self-efficacy for the negative emotions DES and ANG were positive. The correlation indices were low. Extraversion and emotional stability were negatively correlated (*r* = −0.130 **, *p* < 0.01). In addition, extraversion was positively correlated with positive emotions and DES. Emotional stability was positively correlated with DES and ANG. The correlation indices were low, except between extraversion and positive emotions (r = 0.347 **, *p* < 0.01) and between emotional stability and ANG (r = 0.359 **, *p* < 0.01), whose values were close to the mean.

### 4.2. Structural Equation Model

A measurement model was tested by analysing full structural equation models using the full-information maximum-likelihood method. For this method, an estimator was used for robust quantitative variables (MLR) [67]. The two dimensions of emotion regulation and the three factors of emotional self-efficacy acted as latent variables. In contrast, emotional stability and extraversion were the observed variables.

The fit model was evaluated using the chi-square index, comparative fit index (CFI), Tucker–Lewis index (TLI), and standardised root mean square residual (RMSEA). The fit was considered satisfactory with values for the CFI and TLI greater than 0.90 and values for RMSEA and SRMR less than 0.08, depending on the complexity of the model [68]. The criteria described by Kline and Kenny et al. [69,70] were followed. The model had two antecedent variables, three consequent variables, and two mediator variables (Figure 1). It had a satisfactory fit: χ^2^(232) = 647.282, *p* < 0.01; CFI = 0.905; TLI = 0.900; RMSEA = 0.050, 90% CI [.046,.055]; SRMR = 0.055. The factor loadings of the latent variables appear in Figure 2.

The model illustrated a direct, positive, and significant effect of cognitive reappraisal on the three variables of emotional self-efficacy (POS, DES, and ANG). Expressive suppression had a negative effect on self-efficacy for positive emotions and a positive effect on self-efficacy for both negative emotions (DES and ANG).

Emotional stability mediated the relationship between cognitive reappraisal and self-efficacy for negative emotions (DES and ANG) but not self-efficacy for positive emotions (POS). However, extraversion mediated both positive and negative (DES and ANG) self-efficacy. The direct relationships of cognitive reappraisal with the three self-efficacy variables had higher indices than the relationships of mediation.

Expressive suppression had direct, positive relationships with self-efficacy for positive and negative (DES and ANG) emotions and with emotional stability. However, it had a negative relationship with extraversion. In this case, the personality trait variables also mediated the relationship between the antecedent and consequent variables, with slightly higher beta values. The model predicted 28.9% of the variance of self-efficacy for positive emotions, 30.1% of the variance for negative emotions of despair and distress (DES), and 40.6% of the variance for negative emotions of anger and irritation (ANG). Associations of greater than 0.25 between variables indicated medium effects. Associations greater than 0.35 indicated large effects [71].

## 5. Discussion and Conclusions

This study had a twofold aim: (i) to analyse the relationships among emotion regulation, the personality traits of extraversion and emotional stability, and the feeling of efficacy for positive and negative emotions in an adolescent population; and (ii) to verify the mediating role of the personality traits of extraversion and emotional stability in the relationship between emotion regulation and self-efficacy for positive and negative emotions. The model, which represents a new line of exploration, is based on an extensive empirical study. The present study also attempted to overcome the limitations highlighted by Caprara et al. [8] by including the variables of emotion regulation and personality, as well as emotional self-efficacy.

The results lead to the following conclusions. The first conclusion is that the analyses show the positive relationships between cognitive reappraisal and self-efficacy for positive and negative emotions (Hypothesis 1). They also show that expressive suppression has a positive relationship with self-efficacy for negative emotions but a negative relationship with self-efficacy for positive emotions (Hypothesis 2). The analysis thus confirms that cognitive reappraisal increases the perception of self-efficacy for coping with adolescents’ positive and negative emotions. In other words, in complex situations of everyday life, adolescents who are able to apply emotion regulation strategies that focus on the antecedent of the emotional experience can cope more effectively with positive and negative (ANG and DES) emotions. In contrast, focusing on the emotional response, such as expressive suppression, negatively affects the self-efficacy for positive emotions and increases feelings of self-efficacy in the use of negative emotions. Therefore, when faced with an emotional experience, adolescents who are more capable of focusing on antecedents (such as self-control or the search for alternatives) will be able to increase their feelings of emotional self-efficacy and will be more likely to succeed and achieve self-management throughout their lives [3,27]. Experiencing emotional efficacy increases the belief of efficacy and helps cope with situations in a potentially successful manner, which will have a multiplying effect on future situations. Gunzenhauser et al. [10] observed similar findings in a German population. These findings are also consistent with those of Gross and John [9], confirming that strategies of expressive suppression are dysfunctional in terms of the regulation of emotional expression. However, it may initially seem contradictory that expressive suppression should be positively related to negative emotional self-efficacy (DES and ANG). Notably, expressive suppression is an emotion regulation strategy, whereas emotional self-efficacy refers to individuals’ beliefs about their ability to address negative emotions and cope with these emotions suitably. Hence, although expressive suppression may be a maladaptive strategy [43], in school contexts, it may be adaptive because it can attenuate the immediate behavioural response and can help individuals cope with overwhelming negative emotions [42].

The second conclusion relates to Hypothesis 3, aimed at analysing the effects of emotion regulation on self-efficacy for positive and negative emotions. The results show direct, positive relationships between cognitive reappraisal and positive and negative (DES and ANG) emotional self-efficacy. The correlation indices are high for the two dimensions of self-efficacy for negative emotions (DES and ANG). A similar situation appears with expressive suppression, which also has direct relationships with self-efficacy for positive and negative (DES and ANG) emotions. However, in this case, the relationship is weaker. The relationships are direct and positive in all cases, except between expressive suppression and self-efficacy for positive emotions, where the relationships are negative. Hence, emotion regulation strategies that focus on the antecedents of emotional experience enable the management of positive and negative emotions by regulating behaviour to achieve more efficient emotional coping [46]. Expressive suppression reduces self-efficacy for positive emotions [38,48]. In sum, the results support the idea that cognitive reappraisal reduces the intensity of negative emotions and strengthens positive emotions. This situation may also lead to positive emotional feelings [37] and an improvement in interpersonal relationships [7,25,36]. In addition, expressive suppression negatively affects the establishment of smooth, efficient social relations [38,39] and is associated with greater negative affect [38,47].

The third conclusion is that extraversion and emotional stability act as mediating variables [21,54] (Hypothesis 3). Extraversion mediates the relationships between the two types of emotion regulation (cognitive reappraisal and expressive suppression) and positive and negative (DES and ANG) emotions. Emotional stability mediates the relationships between the two types of emotion regulation (cognitive reappraisal and expressive suppression) and self-efficacy for negative emotions (DES and ANG). However, it does not mediate self-efficacy for positive emotions (Hypothesis 4). People who use antecedent-focused emotion regulation strategies and who are emotionally stable will tend to feel more self-efficient in coping with the negative emotions of despair and anger. In summary, emotional stability provides resources to cope with negative emotions more suitably [19]. Even so, it does not seem to have an effect on positive emotions. Positive emotions seem to be more strongly linked to antecedent-focused strategies in emotional experience. One possible explanation is that people with high emotional stability feel that negative emotional coping is an achievement and pay less attention to positive emotional coping [45,46]. The results partially confirm Hypothesis 4 and are consistent with earlier research, although some studies have considered neuroticism instead of emotional stability (given that they are opposites) [45]. In an adult population, scholars have also observed links between neuroticism and negative emotions but not between self-efficacy and positive emotions [45].

Regarding extraversion, the findings corroborate its relationship with positive and negative emotional self-efficacy, as well as confirming its mediating role in the relationships between emotion regulation and positive and negative (DES and ANG) emotional coping. People with a tendency towards extraversion tend to perceive themselves as more effective in coping with positive and negative emotions [52]. This perception may be partly because extroverts easily establish social relationships. Therefore, they have more opportunities to cope with their emotional state and thus manage it efficiently. They also tend to exhibit more positive emotional states and to have experiences involving positive affect [53]. Moreover, extraversion is positively associated with negative emotional self-efficacy (DES), which refers to handling emotions of despair and distress. Faced with highly emotionally charged situations, extroverts are more capable of tackling negative emotional states that cause despair and distress. For instance, a recent study [22] showed links between extraversion and lower levels of physical symptoms associated with negative emotions.

In sum, adolescents with greater extraversion and emotional stability are more likely to cope with highly emotionally charged situations more effectively, particularly those with a negative emotional charge that leads to both externalising and internalising problems such as aggression, anger, distress, or despair. Likewise, the use of antecedent-focused emotion regulation strategies, coupled with emotional stability, can act as a strength that helps adolescents deal with negative emotional burdens. Therefore, they will avoid externalising problems such as aggression or anger, as well as internalising problems such as distress or despair.

## 6. Limitations and Future Research

This study is not free from limitations. First, the results should be taken with some caution, given that this study was cross-sectional. Data gathering was carried out in a single evaluation, which may have been influenced by respondents’ degree of willingness to participate. This study shows the existence of direct relationships, but causal relationships cannot be established. A longitudinal study would strengthen the results. Second, the variables included in this study were part of a more extensive study. Therefore, bias may appear because of participants’ tiredness. Even so, possible tiredness was taken into account, and the sessions were kept short. Third, this study focused on a general Spanish adolescent population (15 to 18 years of age). It would be of interest for future studies to include younger adolescents and, if possible, adolescents from different sociocultural backgrounds. As noted by Kim et al. [24], extraversion and emotional stability may be moderated by culture. Finally, some authors have expressed doubts about the use of short personality questionnaires [72]. However, the TIPI is widely accepted by the research community [57,58,59,60,62].

## Figures and Tables

**Figure 1 behavsci-14-00206-f001:**
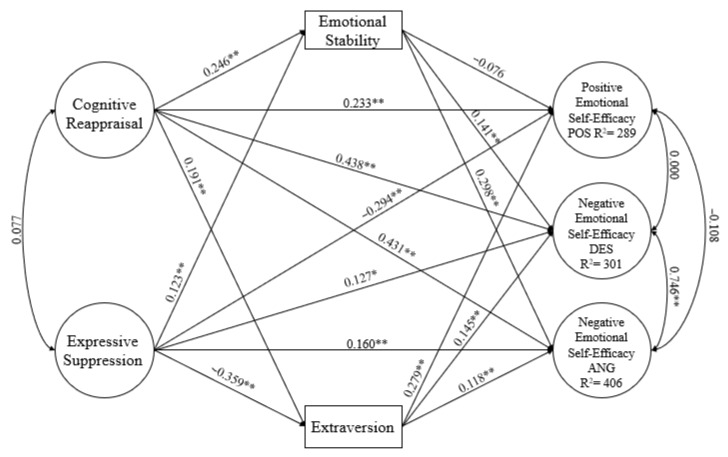
Model with antecedent, consequent, and mediator variables. ** *p* < 0.01; * *p* < 0.05.

**Figure 2 behavsci-14-00206-f002:**
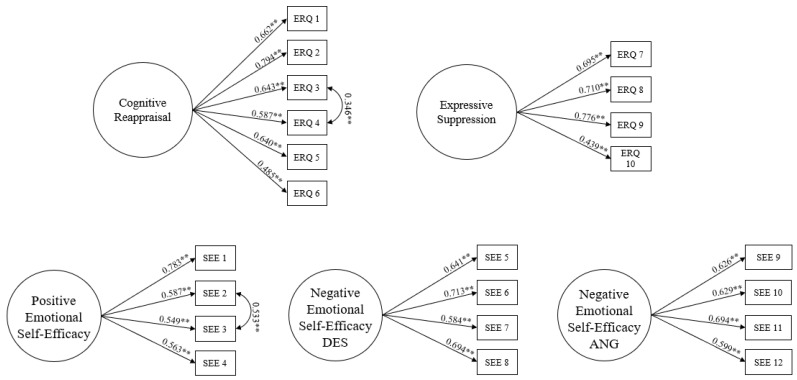
Factor loadings of latent variables. ** *p* < 0.01.

**Table 1 behavsci-14-00206-t001:** Descriptive analyses of the variables.

	Mean	Standard Deviation	Asymmetry	Kurtosis	Minimum	Maximum
Cognitive reappraisal	4.40	0.986	−0.023	−0.512	2.50	6.33
Expressive suppression	3.72	1.26	0.020	−0.957	1.50	6.00
Extraversion	4.89	1.28	−0.129	−0.917	2.50	7.00
Emotional stability	4.29	1.20	−0.114	−0.743	2.00	6.50
Positive emotional self-efficacy (POS)	4.49	0.490	−0.653	−0.821	3.50	5.00
Negative emotional self-efficacy (DES) *	3.17	0.716	0.073	−0.901	2.00	4.50
Negative emotional self-efficacy (ANG) **	2.85	0.772	0.054	−0.883	1.50	4.25

* DES: self-efficacy for coping with despair and distress; ** ANG: self-efficacy for coping with anger and irritation.

**Table 2 behavsci-14-00206-t002:** Correlation analysis.

	1	2	3	4	5	6
1. Cognitive reappraisal	-					
2. Expressive suppression	0.070	-				
3. Extraversion	0.141 **	−0.299 **	-			
4. Emotional stability	0.242 **	0.107 **	−0.130 **	-		
5. Positive emotional self-efficacy (POS)	0.196 **	−0.297 **	0.347 **	−0.061	-	
6. Negative emotional self-efficacy (DES) *	0.408 **	0.103 **	0.128 **	0.236 **	0.109 **	-
7. Negative emotional self-efficacy (ANG) **	0.422 **	0.145 **	0.075	0.359 **	−0.002	0.595 **

** *p* < 0.01; * DES: self-efficacy for coping with despair and distress; ** ANG: self-efficacy for coping with anger and irritation.

## Data Availability

Not available according to Organic Law 3/2018, which regulates the Protection of Personal Data, and Regulation (EU) 2016/679.

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
