# Peer review of "Emotion Regulation and Self-Efficacy: The Mediating Role of Emotional Stability and Extraversion in Adolescence"

_behavsci, 2024, doi:10.3390/bs14030206_

Round 1

Reviewer 1 Report

Comments and Suggestions for Authors

First, thank you for the opportunity to review this work.

In reading it, I found good results but, at the same time, many critical elements. Although the analysis is good, it needs a better theoretical background. The article describes the mediating role of emotional stability and extraversion in adolescents. The authors describe a study with a sample of 703 subjects aged between 15 to 18. 

The authors use a structural equation model where cognitive reappraisal and expressive suppression are the independent variables; I doubt the theoretical model because I am not entirely convinced that emotional regulation strategies can predict personality traits (emotional stability and extraversion). I expect the opposite to be the case, considering that personality traits are stable variables that can be critical predators for behavior and cognitive and emotional strategies. Therefore, the model needs to be modified from a theoretical point of view. Furthermore, I wonder why the authors have only looked at two personality traits and have not considered the others. In addition to these essential aspects, I think it is necessary to consider:

1) The period of data collection should be specified in the methods.

2) A more accurate description of the questionnaires administered is required, such as the number of items and some examples of questions in the instruments (sometimes).

3) Standardize the font of the alphas and choose a uniform style (Alpha or α).

4) It is unclear why the two personality traits are argued within the instruments (TIPI); the author's choice of mediating variables should be placed in other, more appropriate sections.

5) A more detailed statistical analysis is needed on page 8, although it is also in the tables. 

6) Some references are in excess (page 10).

7) The bibliography needs to be formatted according to the journal guidelines.

The manuscript needs to be better grounded to be published.

Author Response

First of all, thank you very much for your review. Your contributions have been well received and he has improved the paper

The theoretical basis has been reorganized and expanded. It has been described why personality traits are unstable and changing in adolescence: lines 30-33 and 65-70.

  1. Data collection period has been added (Page 6).
  2. The number of items has been completed and an example of an item has been added (Pages 5-6).
  3. The alphas presentation style has been standardized (Pages 5-6).
  4. The mediating variables have been explained in the section Aims and Hypotheses (Pages 4 - 5).

Kind regards

Reviewer 2 Report

Comments and Suggestions for Authors

The manuscript provides a valuable contribution to the understanding of emotion regulation, personality traits, and self-efficacy in adolescents. The methodology is sound, and the presentation of results is clear. This manuscript meets the criteria of the journal. In the following, I will outline the main concerns in some detail. 

Major concerns

1. The introduction successfully highlighted the significance of understanding emotion regulation, personality traits, and self-efficacy in adolescents. It mentions relevant existing literature, providing a theoretical foundation for the study. However, the introduction could benefit from a more explicit articulation of the research gap or problem statement that this study aims to address. Clearly defining this gap would strengthen the justification for the current research.

2. The provided text does not explicitly exclude the alternative models (etc., personality serve as independent variables and emotion regulation as the mediating variable; or extraversion and emotional stability moderated the relationship between emotion regulation and self-efficacy). Considering alternative models is a crucial aspect of robust research methodology. Failure to explore other potential models may limit the depth of understanding and could potentially lead to incomplete or biased conclusions. Discussing alternative models would contribute to the overall rigor of the research by addressing potential alternative explanations for the observed relationships. If the author believes that the model used in this study is indeed the most suitable, it is recommended that the author employ theory-driven or data-driven methods to exclude the aforementioned alternative models.

3.The discussion section effectively synthesizes the results, drawing connections between emotion regulation, personality traits, and self-efficacy. The elaboration on cognitive reappraisal and expressive suppression strategies is insightful. However, the discussion would benefit from a deeper exploration of the practical implications of the findings for educators, psychologists, or other stakeholders working with adolescents. Furthermore, the discussion could explicitly link back to the existing literature, reinforcing how the current study contributes to the broader field. For instance, this study's findings have implications for enhancing the emotional efficacy among the adolescent population and how it can be applied in interventions addressing emotional issues in individuals with different personality traits.

4. The two paragraphs starting from line 369 discussed the mediating role of extraversion and emotional stability. However, some of the descriptions may give readers the impression that extraversion and emotional stability are moderators. Please enhance the accuracy of the expression in these two paragraphs.

Minor concerns

1.There is an error on line 420; "5" should be corrected to "6."

2.The description of instruments is clear. It's beneficial to include validity coefficients for the TIPI, even though they are widely accepted and observable variables (not latent variables).

Comments on the Quality of English Language

1.Consider the use of transitional phrases to improve the flow between sentences.

2. The sentences are well-constructed for the most part, but there are instances where sentence structure could be simplified for clarity. Consider breaking down complex sentences into shorter ones to improve readability.

Author Response

Thank you very much for your review. Your contributions have been well received and he has improved the paper

  1. The introduction has been reorganized and the research gap explained (Pages 1–4).
  2. The psychometric analyzes presented are only aimed at offering the reader the stability of the latent variables in the hypothesized model. For this reason, it has not been considered appropriate to offer an analysis of alternative models. Alternative models are not objectives of this research, although the idea is very good and will be considered in future research (Pages 8-9).
  3. The Discussion and conclusions section (Pages 9-11) has been reorganized.
  4. Latent and observed variables have been completed (Page 7).
  5. Fixed bug 6 (instead of 5).
  6. The item-total correlation coefficient (in this case item-item) has been added according to Aiken, L. R. Three coefficients for analyzing the reliability and validity of ratings. Educational and Psychological Measurement 1985, 45, 131-142. https://doi.org/10.1177/0013164485451012
  7. The entire text has been revised seeking a more simplified structure in order to improve readability.

Kind regards

Reviewer 3 Report

Comments and Suggestions for Authors

Dear Authors, 

I was pleased to read your paper about emotion regulation and self-efficacy in adolescence and young adulthood. 

However, I have some comments about the manuscript:

  • Introduction: ok for me
  • Hypotheses: It is not clear to me why you stated hypotheses that are robust evidence in the scientific literature; "Emotion regulation has a direct effect on self-efficacy for positive emotions (POS) and negative 180 ones (DES and ANG), following the recent study by [39]" this is not a hypothesis, this is a piece of evidence that you data replicate (or not). Where is the novelty? Same for H4. I suggest revising the hypothesis to emphasize ones that add novelty and relevance to the scientific community.
  • Methods: ok for me
  • Discussion: this must be improved considering the hypotheses reformulation. Additionally, I recommend that the issue of the model's explanation of only 30% of the variance be addressed more effectively. This result may not be considered satisfactory, and therefore, it should be highlighted in the limitation paragraph. Perhaps, the choice of the instrument selected for the investigation could help in explaining the model's weakness; in any case, I guess that this should be discussed (why use a 10-item scale for personality if this psychological construct is the focus of your study?)
Comments on the Quality of English Language

As a non-native English speaker, I find the text to be awkwardly phrased in certain places. The text should be revised to ensure readability.

Author Response

Thank you very much for your review. Your contributions have been well received and he has improved the paper.

Hypotheses have been rearranged.

References are eliminated because the investigations are in the adult population. This research is in adolescence (Page 5).

The Discussion section (Pages 9-11) has been reorganized.

Limitations include the instrument that assesses personality (TIPI) (Page 11).

The entire text has been reviewed and corrected, seeking a more simplified structure to improve readability.

Kind regards

Reviewer 4 Report

Comments and Suggestions for Authors

Dear authors, this is a well-done article. The only thing I would recommend is to improve your section "Discussion and conclusions". It is rich in discussion but poor in conclusions. Maybe you refer to questions like "What does that mean for e.g. educators, employers, policymakers etc...." . Perhaps you can guide relevant stakeholders in how to make practical use of your findings.

Else, well done!

Author Response

Thank you very much for your review. Your contributions have been well received and he has improved the paper.

The Discussion and Conclusion section (Pages 9-11) has been reviewed and reorganized.

Kind regards

Round 2

Reviewer 1 Report

Comments and Suggestions for Authors

Dear authors, 

I appreciate your effort in reviewing your article on emotion regulation and self-efficacy in adolescence. In reading it, I found the introduction, instruments, statistical analysis, discussion, and conclusion much more detailed and described.

Kind regards